# Additional Anterolateral Ligament Reconstruction Helps Patients Improve Dynamic Postural Stability in Revision Anterior Cruciate Ligament Reconstruction

**DOI:** 10.3390/medicina59071242

**Published:** 2023-07-03

**Authors:** Joon Kyu Lee, Seung-Ik Cho, Dhong-Won Lee, Sang-Jin Yang, Tae-Wook Kim, Jin-Goo Kim

**Affiliations:** 1Department of Orthopaedic Surgery, Konkuk University Medical Center, Research Institute of Medical Science, Konkuk University School of Medicine, Seoul 05030, Republic of Korea; ndfi@naver.com; 2Department of Orthopaedic Surgery, Konkuk University Medical Center, Seoul 05030, Republic of Korea; bestjjo@naver.com (S.-I.C.); wonbayo@naver.com (D.-W.L.); 20200134@kuh.ac.kr (T.-W.K.); 3Department of Health & Exercise Management, Tongwon University, Gwangju-si 12813, Republic of Korea; sjyang@tw.ac.kr; 4Department of Orthopaedic Surgery, Myong-Ji Hospital, Goyang-si 10475, Republic of Korea

**Keywords:** dynamic postural stability, anterolateral ligament reconstruction, revision anterior cruciate ligament reconstruction, Y-balance test

## Abstract

*Background and Objectives*: The goal in treating anterior cruciate ligament (ACL) injury especially in revision cases is return to sports activity by regaining dynamic postural stability. Among various methods to achieve this goal, additional anterolateral ligament reconstruction (ALLR) is gaining attention. The purpose of this study was to evaluate the effects of additional ALLR in revision ACL reconstruction (RACLR). *Materials and Methods*: Patients who underwent RACLR between July 2015 and June 2018 were enrolled. The exclusion criteria were less than 1-year follow-up, age older than 45 years, concomitant multiple ligament injuries, contralateral knee injury, subtotal or total meniscectomized state, and articular cartilage lesions worse than Outerbridge grade 3. Thirty-nine patients (20 patients; RACLR only (Group A), 19 patients; RACLR with additional ALLR (Group B)) were included. Clinical scores (Lysholm score, subjective International Knee Documentation Committee (IKDC) score, Tegner activity scale), isokinetic strength test, single-leg-hop for distance test (SLHDT), Y-balance test (YBT) were checked preoperatively and 1-year postoperatively. *Results*: Limb symmetry index values in YBT showed significantly better result in Group B 1-year postoperatively (Group A: 97.2 ± 4.0, Group B: 100.3 ± 2.9, *p* = 0.010), although there were no differences preoperatively between groups (Group A: 90.4 ± 6.7, Group B: 89.3 ± 5.5, *p* = 0.594). Regarding clinical scores, isokinetic strength tests, and SLHDT, there were no differences between groups preoperatively nor 1-year postoperatively. *Conclusions*: Additional ALLR in RACLR helped patients gain better dynamic postural stability at 1-year postoperative follow-up.

## 1. Introduction

Anterior cruciate ligament (ACL) injury is common in sports activities. The annual incidence of ACL injuries worldwide is approximately 2 million cases [1]. Even though most studies reported high success rates for ACL reconstruction (ACLR), there have also been numerous reports regarding the failure of ACLR. Among the many factors contributing to the failure, residual rotational laxity is considered to be one of the most critical [2,3,4]. Recently, anterolateral ligament (ALL) reconstruction (ALLR) has emerged as an effective method to provide rotational stability in ACLR, especially in revision ACLR cases [5,6,7,8,9]. Furthermore, the improved rotational stability could protect the ACL graft from excessive load stress during rehabilitation and recovery [6]. Vincent et al. and Claes et al. each described detailed anatomy of the ALL and provided grounds for ALLR [10,11]. A number of cadaveric studies demonstrated the biomechanical basis for improved rotational stability by adding ALLR [12,13,14].

The ultimate goal of ACLR, whether it is a primary or revision case, is to return to the level of sports activity prior to the injury [15]. To achieve this goal, regaining dynamic postural stability of the affected limb, thus improving asymmetrical control in single-leg balance has been reported to be important [16,17]. Therefore, it is imperative to assess the effect of additional ALLR with regard to dynamic postural stability especially in revision ACLR cases, to justify the procedure. The Y-balance test (YBT), modified from the star excursion balance test (SEBT), is a popular test to assess dynamic postural stability in clinical settings. It is used to assess performance during single-leg balance task. While performing the YBT, the subjects are required to stand on one leg stance and squat down while pushing a sliding plastic reach indicator as far as they can. This movement requires concentric contraction strength of the quadriceps muscles and eccentric contraction strength of the hamstring muscles. Moreover, it requires the ability to maintain balance, which reflects the patient’s proprioceptive function [18,19]. The strength and balance of the involved knee are key to perform well in YBT [20,21,22].

The purpose of this study was to evaluate the effects of additional ALLR in revision ACLR, especially regarding dynamic postural stability. The hypothesis was that additional ALLR in revision ACLR would improve dynamic postural stability compared to that in isolated revision ACLR cases.

## 2. Materials and Methods

### 2.1. Patient Selection

Fifty-three consecutive patients who underwent revision ACLR by a single surgeon (*) between July 2015 and June 2018 were included in this study. All patients were confirmed to have primary ACL graft rupture during the surgery. The exclusion criteria were as follows: (1) less than 1-year follow up, (2) age older than 45 years, (3) concomitant multiple ligament injuries, including posterolateral corner injury, (4) contralateral knee injury, (5) a subtotal or total meniscectomized state, and (6) articular cartilage lesions with worse than modified Outerbridge grade 3. Among the 53 patients, fourteen met at least one of the exclusion criteria. A total of 39 patients were included in this study. Twenty patients underwent revision ACLR only from July 2015 to December 2016 (Group A) and 19 patients underwent revision ACLR with additional ALLR from January 2017 to June 2018 (Group B) (Figure 1). There were no significant differences in demographic data between the groups (Table 1). This retrospective study was conducted in accordance with the Declaration of Helsinki, and approved by the institutional review board of Konkuk University Medical Center (KUMC 2020-03-069). Informed consent was obtained from all enrolled patients either directly at follow-up office visits or over the phone.

### 2.2. Surgical Technique

Revision ACLR and additional ALLR were performed as previously described [6]. Here are the brief explanations of the procedures.

#### 2.2.1. Revision ACLR

A tibialis anterior tendon allograft (fresh frozen, Korea Bone Bank, Seoul, Republic of Korea) was chosen as the graft. After identifying the primary ACL graft tear, the graft was removed. A new femoral tunnel was usually created at a more posterior and lateral position than the initial femoral tunnel position using the outside-in method with FlipCutter (Arthrex, Naples, FL, USA). A 9-mm diameter graft was prepared and inserted. The ACL TightRope (Arthrex) was used for femoral fixation, and after distal pulling of the graft for tensioning, a bio-interference screw (Matryx; Conmed Linvatec, Largo, FL, USA) and staple were used for tibial fixation.

#### 2.2.2. ALLR

A gracilis tendon allograft (fresh frozen, Korea Bone Bank, Seoul, Republic of Korea) was chosen as the graft. An additional femoral tunnel for ALL was created just proximal and posterior to the lateral femoral epicondyle using a cannulated reamer with a 25 mm depth. An additional tibial tunnel for ALL was created at approximately 10 mm below the joint line between the Gerdy’s tubercle and the fibular head with a 25 mm depth. A 6-mm diameter graft was prepared and inserted into both tunnels by the pull-out method. Bio-interference screws (Matryx), 7 mm in diameter, were used for femoral and tibial fixation. Femoral fixation was performed first, and tibial fixation was performed at 30° in knee flexion and neutral rotation.

### 2.3. Postoperative Rehabilitation

Both groups followed the same postoperative rehabilitation protocol [6]. Weight bearing was permitted as tolerated immediately after surgery while wearing an ACL brace (Legend; DonJoy, Lewisville, TX, USA) locked in full extension. Range of motion exercises for the knee was permitted 3 weeks after surgery, allowing for flexion between 0° and 90° while wearing an ACL brace. Gradual increase in the flexion angle was encouraged with the goal of achieving full flexion six weeks after surgery. The brace was worn for a full two months. At 6 weeks post-surgery, closed kinetic chain exercises, such as squats and single-leg balancing, were initiated. Open kinetic chain exercises were initiated 2 months post-surgery. At 6 months post-surgery, running was permitted after verifying the recovery of the muscle power in the operated leg. At 9 months post-surgery, return-to-competitive sports activity was allowed.

### 2.4. Outcome Assessment

The outcome assessments were performed by the assistant trainers who were blind to the type of surgery patients received.

#### 2.4.1. Subjective Knee Scores

Three knee scores (Lysholm score, subjective International Knee Documentation Committee (IKDC) score, and Tegner activity scale) were obtained preoperatively and 1-year postoperatively. The scores were managed by an independent research investigator.

#### 2.4.2. Isokinetic Muscle Strength Test

All patients underwent isokinetic muscle strength test preoperatively and 1-year postoperatively. A Biodex system IV dynamometer (Biodex Medical Systems, Shirley, NY, USA) was used. Patients were instructed on how to operate the isokinetic testing equipment and were given two trials before the actual assessment. All tests were performed by highly trained exercise specialists. The testing protocol was 60°/s for both knee extension and flexion. The deficit (%) of the affected side compared to the unaffected side was used for evaluation.

#### 2.4.3. Single Leg Hop for Distance Test

All patients underwent single leg hop for distance (SLHD) test at 1 year postoperatively. Patients were asked to jump forward as far as possible with one foot. They were given three attempts, and the longest distances for the affected and unaffected sides were measured. The limb symmetry index (LSI, %) was calculated.

#### 2.4.4. Y-Balance Test

The YBT kit (Figure 2) was used for evaluation. Patients performed anterior, posteromedial, and posterolateral reach. Patients were requested to put both hands on the same side of the waist (iliac crest) during the test. The score was not recorded in cases when the patient was not able to maintain a single-legged stance on the platform while performing the test and in cases when the patient’s hands were removed from the waist. All patients completed three trials in each direction and recorded the best score. To avoid patients’ fatigue, sufficient rest time was provided between the trials. The true leg length, which is the distance from the anterior superior iliac spine to the medial malleolus, was measured in all patients. All measuring procedures were performed by trained exercise specialists. The YBT composite score was calculated using the following formula: [(anterior + posteromedial + posterolateral)/(3 × Leg Length)] × 100
anterior, posteromedial, posterolateral, leg length unit: cm.

Then, the LSI (%) was calculated. The sub-portions of the YBT score (anterior reach score, posteromedial reach score, posterolateral reach score, respectively) analyses were also performed.

### 2.5. Statistical Analysis

All statistical analyses were performed using SPSS program for Windows version 21.0 (SPSS, Chicago, IL, USA). The Shapiro-Wilk test was used to check the data for normality. Comparisons between the groups were performed using the Student *t* test for continuous normal distribution data and the Mann-Whitney U test for ordinal categorical and non-normal distribution data. Comparisons within the groups between the preoperative and postoperative data were performed using a paired *t*-test for continuous normal distribution data and a Wilcoxon signed-rank test for ordinal categorical and non-normal distribution data. A Pearson chi-square test was used for nominal categorical data. The level of significance was set at *p* ≤ 0.05. A post-hoc power analysis was performed using G*Power version 3.1.2 to assess the validity of the sample size based on the comparison of postoperative 1 year YBT composite score LSI.

## 3. Results

### 3.1. Subjective Knee Scores

The Lysholm and IKDC subjective scores improved significantly 1 year after revision ACLR in both groups (all *p* < 0.05), and the Tegner activity score recovered to the preoperative level in both groups (Table 2). There were no significant differences between the groups in all scores either preoperatively or 1 year postoperatively (all *p* > 0.05), although group B showed better 1-year postoperative Tegner activity scale (group A: 6.9, group B: 7.6) (Table 2).

### 3.2. Isokinetic Muscle Strength Test

The knee extensor and flexor muscle strengths compared to the unaffected side improved significantly 1 year after revision ACLR in both groups (all *p* < 0.05) (Table 3). There were no significant differences between the groups, either preoperatively or 1-year postoperatively (all *p* > 0.05) (Table 3).

### 3.3. Single Leg Hop for Distance Test

There was no significant difference in the postoperative 1-year SLHD test results between the groups (*p* = 0.715) (Table 4).

### 3.4. Y-Balance Test

The LSIs of the scores improved significantly 1 year after revision ACLR in both groups (all *p* < 0.05) (Table 5). There were no significant differences in the preoperative LSIs of the scores between the groups (all *p* > 0.05, composite score, anterior reach score, posteromedial reach score, and posterolateral reach score, respectively). Group B showed significantly better composite scores LSI 1-year postoperatively (*p* = 0.01). All of the Group B sub-portion scores were better than Group A scores 1-year postoperatively, with the posteromedial reach score showing the most difference (Table 5).

### 3.5. Power of the Study

A post hoc power analysis showed that the sample size of 39 patients (20 Group A and 19 Group B) revealed sufficient statistical power (0.99) based on the comparison of postoperative 1 year YBT composite score LSI.

## 4. Discussion

The important finding of the present study was that additional ALLR helped gain better dynamic postural stability evaluated by YBT in revision ACLR cases at 1-year postoperative follow-up. 

The renewed attention and confirmation of the ALL anatomy was made by several investigators in the early 2010s. Vincent et al. [11] identified the ALL in patients undergoing total knee arthroplasty and Clae et al. [10] also pointed out the ligament in the cadaver specimen. Many biomechanical studies of ALL followed and revealed that ALL is an important structure for the anterolateral rotatory stability of the knee. In a cadaver study, Sonnery-Cottet et al. showed that ALL affected rotational control of the knee at varying degrees of knee flexion during a pivot shift maneuver [23]. Parsons et al. reported in their cadaver study that ALL is an important stabilizer of the knee’s internal rotation at flexion angles greater than 35° [13]. Nielsen et al. also reported that augmented ALLR with ACLR in a cadaveric setting reduced internal rotation, varus rotation, and anterior translation knee laxity [12].

There have been several reports suggesting that the clinical result of revision ACLR is generally inferior to that of primary ACLR [24,25]. In the case of revision ACLR, since it is a second or more relevant surgery on one knee, there is lesser room to spare than in primary ACLR. Therefore, close attention should be paid to the cause of the failure and the presenting symptoms. Both the cause and the symptoms should be managed thoroughly to prevent re-rupture. Residual rotational laxity is known to be the major cause of failure after ACLR, and a high pivot shift is one of the major symptoms. Recently, numerous studies reported significantly better knee stability after revision ACLR with additional ALLR. Yoon et al. reported that additional ALLR in revision ACLR cases with high-grade pivot shift improved both anteroposterior stability and rotational stability [7]. Louis et al. reported improved rotational stability and re-rupture risk with additional ALL stabilization procedures in revision ACLR cases in their multi-center study [5]. However, in terms of subjective patient outcomes usually evaluated using clinical questionnaires and scores, controversy persists regarding the effect of additional ALLR in revision ACLR cases. Yoon et al. reported no difference in clinical scores, while Lee et al. reported significantly better scores [6,7]. Furthermore, these previous studies concentrated on traditional clinical outcomes and stability assessments [5,7].

Among the various measures that evaluate the results of either ACLR or revision ACLR, postural stability is an important measure to be taken seriously. Lack of sufficient postural stability of the involved limb is considered the main factor in the failure of ACLR. Paterno et al. reported that postural instability after ACLR is a predictor for ACL re-injury [26]. Whether the increased knee stability by additional ALLR in revision ACL cases improves the postural stability of the patient has not been studied widely.

Postural stability generally means the ability to regain the balance or control of the trunk and the lower limb [21]. Postural stability can be assessed in two different ways: static and dynamic [27,28]. In static postural stability evaluation, the subject is required to establish a firm base of support and maintain the position while limiting body movement. In dynamic postural stability evaluation, the subject is required to maintain balance while moving from a dynamic to a static position. For the general assessment after ACLR, such as evaluating the duration of rehabilitation and when to start sports activity, dynamic postural stability evaluation is more appropriate than static postural stability evaluation. Head et al. insisted that dynamic postural stability should be assessed carefully in the return-to-sports decision-making process after ACLR [27,29].

The SEBT developed by Gray et al. was considered a reliable and valid method to measure dynamic postural stability [18]. However, the test was too time-consuming in a clinical setting. To apply the SEBT practically, the YBT was developed and its reliability has been proven by many researchers [22,30]. YBT is a relatively inexpensive and easy to apply test for clinicians [31]. While performing the YBT, the subjects are required to stand on one leg stance and squat down as far as they can. This movement requires concentric contraction strength of the quadriceps muscles and eccentric contraction strength of the hamstring muscles. Moreover, it requires the ability to maintain balance, which reflects the patient’s proprioceptive function [18]. Nowadays, YBT is considered one of the most popular research tools used for the assessment of dynamic postural stability [21]. Several studies have used YBT as one of their evaluation measures for ACLR results. However, few studies have used YBT in revision ACLR cases. In this study, we placed emphasis on the dynamic postural stability evaluated using YBT in comparing revision ACLR results for additional ALLR and confirmed that additional ALLR indeed improves dynamic postural stability in revision ACLR cases.

This study has limitations. First, this study was a retrospective comparative study. A prospective randomized study is always desirable in such kinds of comparative clinical analyses; however, it is difficult to design a prospective randomized study in revision ACLR cases because patients are usually desperate for the success of the surgery and do not want to participate in any kind of clinical study. Second, postoperative 1-year evaluation might be considered too short for the evaluation of results. However, return-to-sports activity is one of the major decision-making processes, and dynamic postural stability is the main factor to consider for those decisions. Because the time to return to sports is usually around a year after surgery, we thought that it is important to perform the evaluation 1-year postoperatively. Lastly, a post hoc power analysis revealed that the sample size of 39 patients showed sufficient statistical power, but it was still low for the comparison study.

## 5. Conclusions

Additional ALLR in revision ACLR helped patients gain better dynamic postural stability evaluated by YBT in revision ACLR cases at 1 year postoperative follow-up.

## Figures and Tables

**Figure 1 medicina-59-01242-f001:**
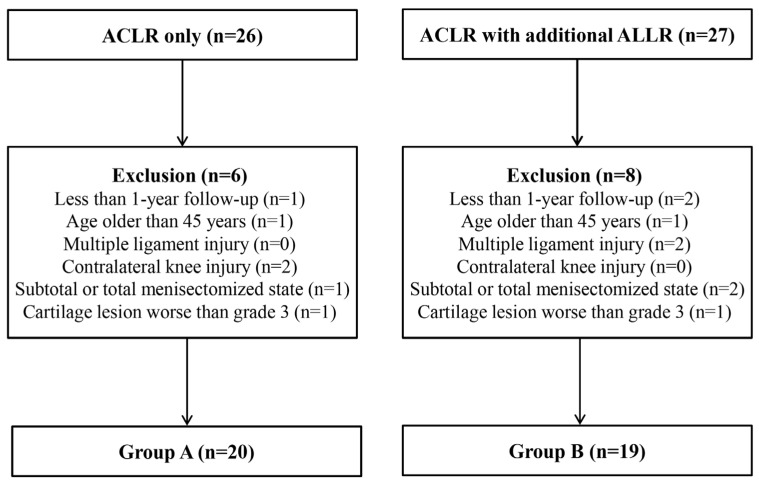
Flowchart of the patient selection process. ACLR; anterior cruciate ligament reconstruction, ALLR; anterolateral ligament reconstruction.

**Figure 2 medicina-59-01242-f002:**
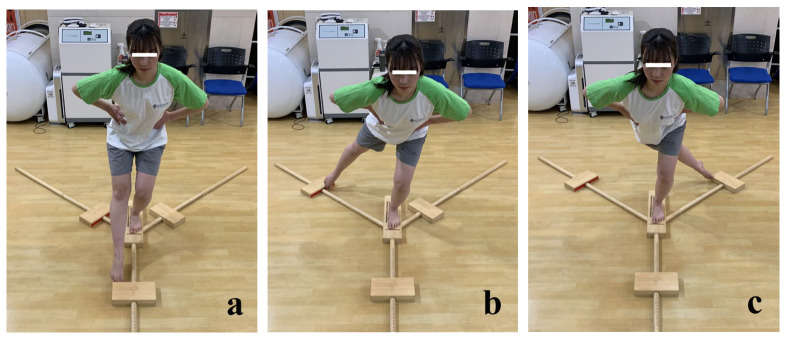
Y-balance test. (**a**) anterior reach direction; (**b**) posteromedial reach direction; (**c**) posterolateral reach direction.

**Table 1 medicina-59-01242-t001:** Comparison of pre-operative demographics between the groups.

	Group A (*n* = 20)	Group B (*n* = 19)	*p* Value
Gender (Male/Female)	16/4	16/3	0.732 ^a^
Age * (years)	25.5 ± 7.87	27.3 ± 10.07	0.642 ^b^
Body Mass Index * (kg/m^2^)	26.3 ± 5.2	25.5 ± 3.1	0.955 ^b^
Involved leg (Right/Left)	8/12	10/9	0.429 ^a^

^a^: Chi square test, ^b^: Mann-Whitney test, *: The values are given as the mean and the standard deviation.

**Table 2 medicina-59-01242-t002:** Comparison of the subjective knee scores between the groups.

	Group A (*n* = 20)	Group B (*n* = 19)	*p* Value
Lysholm score *			
Preoperative	73.1 ± 9.8	71.9 ± 13.0	0.755 ^a^
Postoperative 1 year	91.4 ± 10.7	91.4 ± 10.9	0.678 ^b^
*p* value	<0.001 ^c^	0.004 ^c^	
Tegner activity score *			
Preoperative	6.9 ± 2.6	6.8 ± 2.2	0.989 ^b^
Postoperative 1 year	6.9 ± 2.0	7.6 ± 1.8	0.196 ^b^
*p* value	0.791 ^c^	0.092 ^c^	
IKDC subjective score *			
Preoperative	65.0 ± 12.9	68.3 ± 10.7	0.383 ^a^
Postoperative 1 year	88.4 ± 12.7	90.5 ± 11.6	0.581 ^b^
*p* value	<0.001 ^c^	<0.001 ^c^	

^a^: Student *t* test, ^b^: Mann-Whitney test, c: Wilcoxon signed-rank test, *: The values are given as the mean and the standard deviation, Abbreviations: IKDC, International Knee Documentation Committee.

**Table 3 medicina-59-01242-t003:** Comparison of the knee muscle strength between the groups.

	Group A (*n* = 20)	Group B (*n* = 19)	*p* Value
Knee extensor strength deficit compared to the contralateral limb (%, 60°/s Biodex dynamometer) *			
Preoperative	25.7 ± 23.1	27.6 ± 18.3	0.779 ^a^
Postoperative 1 year	14.5 ± 15.3	11.2 ± 11.8	0.463 ^a^
*p* value	0.008 ^c^	0.002 ^c^	
Knee flexor strength deficit compared to the contralateral limb (%, 60°/s Biodex dynamometer) *			
Preoperative	18.8 ± 17.1	20.4 ± 21.7	0.807 ^a^
Postoperative 1 year	0.2 ± 27.4	9.0 ± 12.5	0.273 ^b^
*p* value	0.006 ^c^	0.035 ^d^	

^a^: Student *t* test, ^b^: Mann-Whitney test, ^c^: Wilcoxon signed-rank test, *: The values are given as the mean and the standard deviation.

**Table 4 medicina-59-01242-t004:** Comparison of the postoperative 1-year single leg hop for distance test.

	Group A (*n* = 20)	Group B (*n* = 19)	*p* Value
Limb symmetry index (%) *	88.2 ± 13.5	89.9 ± 12.7	0.715 ^a^

^a^: Mann-Whitney test, *: The values are given as the mean and the standard deviation.

**Table 5 medicina-59-01242-t005:** Comparison of the Y-balance test between the groups.

	Group A (*n* = 20)	Group B (*n* = 19)	*p* Value
YBT composite score LSI (%) *			
Preoperative	90.4 ± 6.7	89.3 ± 5.5	0.594 ^a^
Postoperative 1 year	97.2 ± 4.0	100.3 ± 2.9	0.010 ^a^
*p* value	<0.001 ^c^	<0.001 ^c^	
YBT anterior reach score LSI (%) *			
Preoperative	87.3 ± 4.8	87.8 ± 13.3	0.856 ^a^
Postoperative 1 year	96.0 ± 7.4	99.7 ± 6.5	0.105 ^a^
*p* value	<0.001 ^c^	0.001 ^c^	
YBT posteromedial reach score LSI (%) *			
Preoperative	91.1 ± 11.4	91.4 ± 6.5	0.383 ^b^
Postoperative 1 year	97.6 ± 4.2	100.2 ± 4.3	0.063 ^a^
*p* value	0.008 ^d^	0.001 ^d^	
YBT score posterolateral portion LSI (%) *			
Preoperative	91.9 ± 9.4	88.3 ± 6.5	0.175 ^a^
Postoperative 1 year	97.7 ± 6.5	100.9 ± 4.3	0.581 ^b^
*p* value	0.010 ^d^	<0.001 ^c^	

^a^: Student *t* test, ^b^: Mann-Whitney test, ^c^: Paired *t* test, ^d^: Wilcoxon signed-rank test, *: The values are given as the mean and the standard deviation. Abbreviations: YBT, Y-balance test; LSI, Limb symmetry index.

## Data Availability

The datasets analyzed in this study are not publicly available but are available from the corresponding author on appropriate request.

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
