# Peer review of "Additional Anterolateral Ligament Reconstruction Helps Patients Improve Dynamic Postural Stability in Revision Anterior Cruciate Ligament Reconstruction"

_medicina, 2023, doi:10.3390/medicina59071242_

Round 1
Reviewer 1 Report
Thanks for the invitation. The purpose of this study was to evaluate the effects of additional ALLR in revision ACL reconstruction (RACLR), and the author found ALLR in RACLR helped patients gain better dynamic postural stability at 1-year postoperative follow-up. I just have several suggestions:
1. Abstract part should be edited. Fewer methods and more results. Some negative results should be mentioned in the Abstract part.
2. As a limitation, low sample size should be mentioned in the discussion part.
the quality of English is good.
Author Response
- Abstract part should be edited. Fewer methods and more results. Some negative results should be mentioned in the Abstract part.
Reply: We have shortened the Methods section and included the negative preoperative Limb symmetry index value in the Y Balance Test, as recommended.
- As a limitation, low sample size should be mentioned in the discussion part.
Reply: We added this matter in the discussion part.
Reviewer 2 Report
Thank you for the opportunity to review the manuscript titled „Additional anterolateral ligament reconstruction helps patients improve dynamic postural stability in revision anterior cruciate ligament reconstruction.” The aim of the study was to o evaluate the effects of additional anterolateral ligament reconstruction (ALLR) in revision ACL reconstruction (RACLR) . Thirty-nine patients were included (20 in RACLR only (Group A) and 19 patients RACLR with additional ALLR (Group B). Based on the study results, the authors conclude that additional ALLR in RACLR helped patients gain better dynamic postural stability at 1-year postoperative follow-up.
The manuscript deals with an important topic due to the frequent injuries of the anterior cruciate ligament. Overall it is well written and needs minor revision in my opinion.
It should be explained who and on what basis decided whether the patient will have only ACLR or ACLR with an additional ALLR. This is important from an ethical point of view.
If the study was retrospective, when were written consents obtained from patients?
The description of the physiotherapy used is too enigmatic. This subsection should be described with more details.
If the examination was performed retrospectively, I understand that all tests performed before the procedurę (preoperative) are a standard procedure that is assessed before ACL reconstruction?
Author Response
It should be explained who and on what basis decided whether the patient will have only ACLR or ACLR with an additional ALLR. This is important from an ethical point of view.
Reply: The time period was the only basis for deciding this. During the study time period, only RACLR was done before 2017 and after January 2017, all the RACLR was done with additional ALLR.
If the study was retrospective, when were written consents obtained from patients?
Reply: During the follow-up office visits.
The description of the physiotherapy used is too enigmatic. This subsection should be described with more details.
Reply: We revised the rehabilitation section with more details as recommended.
If the examination was performed retrospectively, I understand that all tests performed before the procedurę (preoperative) are a standard procedure that is assessed before ACL reconstruction?
Reply: Yes, we have one of the leading knee sports injury centers in Korea, and we have established standard preoperative assessment procedures for a long time.
Reviewer 3 Report
This manuscript covers a current issue. It is well organized and subscribes to the highest standards of how to report a research study. I find no reason to suggest any additions or corrections. Thank you for allowing me to review this fine manuscript.
Title: Additional anterolateral ligament reconstruction helps patients improve dynamic postural stability in revision anterior cruciate ligament reconstruction.
Abbreviations:
anterolateral ligament (ALL) reconstruction (ALLR)
anterior cruciate ligament (ACL) injury
anterolateral ligament reconstruction (ALLR)
ACL reconstruction (ACLR)
Y-balance test (YBT)
star excursion balance test (SEBT)
ABSTRACT: Clearly written and thorough. Presents the results fairly.
Purpose:
The goal in treating anterior cruciate ligament (ACL) injury especially in revision cases is return to sports activity by regaining dynamic postural stability. The purpose of this study was to evaluate the effects of added ALLR in revision ACL reconstruction (RACLR).
Introduction:
Recently, anterolateral ligament (ALL) reconstruction (ALLR) has emerged as an effective method to provide rotational stability in ACLR, especially in revision ACLR cases [5-9].
it is imperative to assess the effect of added ALLR regarding dynamic postural stability especially in revision ACLR cases, to justify the procedure. The Y-balance test (YBT), modified from the star excursion balance test (SEBT), is a popular test to assess dynamic postural stability in clinical settings.
Methods:
This was a retrospective comparative study. Patients who underwent RACLR between 22 July 2015 and June 2018 were enrolled.
Fifty-three consecutive patients who underwent revision ACLR by a single surgeon between July 2015 and June 2018 were included in the study. A total of patients were included in this study. Twenty patients underwent revision ACLR only from July 2015 to December 2016; (Group A) and patients underwent revision ACLR with additional ALLR from January 2017 to June 2018 (Group B).
Informed consent was obtained from all enrolled patients.
A post hoc power analysis showed that the sample size of 39 patients (20 Group A and 19 Group B) revealed sufficient statistical power (0.99) based on the comparison of postoperative 1 year YBT composite score LSI.
Methods:
Research Design:
Twenty patients underwent revision ACLR only from July 2015 to December 2016 81 (Group A) and 19 patients underwent revision ACLR with additional ALLR from Janu-82 ary 2017 to June 2018 (Group B).
Revision ACLR: A tibialis anterior tendon allograft (fresh frozen, Korea Bone Bank) was chosen as graft.
ALLR: A gracilis tendon allograft (fresh frozen, Korea Bone Bank) was chosen as the graft.
Postoperative rehabilitation: Both groups followed the same postoperative rehabilitation protocol.
Two groups, A and B. Outcome assessment: The outcome assessments were performed by the assistant trainers who were blind to the type of surgery patients received. Evaluation tests are well described. Some tests were preoperative and postoperative. Some test were applied Immediate post operative and at 1 year post operative; other tests were given only at 1 year post operative.
Statistics: Excellent description of statistical tests performed, when, and why.
Results:
Conclusion:
The important finding of the present study was that additional ALLR helped gain better dynamic postural stability evaluated by YBT in revision ACLR cases at 1-year post-operative follow-up.
Discussion:
Very good discussion including history of ALL, ALLR and reasoning for this study.
Figures:
Fig 1: Excellent & Necessary.
Fig 2: Excellent & Necessary.
Table 1. Excellent & Necessary.
Table 2. Excellent & Necessary
Table 3. Excellent & Necessary
Table 4. Excellent & Necessary
Table 5. Excellent & Necessary

Author Response
This manuscript covers a current issue. It is well organized and subscribes to the highest standards of how to report a research study. I find no reason to suggest any additions or corrections. Thank you for allowing me to review this fine manuscript.
Reply: Thank you very much for the review of our manuscript. It was encouraging to all the authors.